# Octa-coordinated alkaline earth metal–dinitrogen complexes M(N$_2$)$_8$ (M=Ca, Sr, Ba)

Qian Wang[1,4], Sudip Pan [2,4], Shujun Lei[1], Jiaye Jin [1], Guohai Deng[1], Guanjun Wang[1], Lili Zhao[2], Mingfei Zhou[1] & Gernot Frenking [2,3]

We report the isolation and spectroscopic identification of the eight-coordinated alkaline earth metal–dinitrogen complexes M(N$_2$)$_8$ (M=Ca, Sr, Ba) possessing cubic ($O_h$) symmetry in a low-temperature neon matrix. The analysis of the electronic structure reveals that the metal-N$_2$ bonds are mainly due to [M(d$_\pi$)]→(N$_2$)$_8$ $\pi$ backdonation, which explains the observed large red-shift in N-N stretching frequencies. The adducts M(N$_2$)$_8$ have a triplet ($^3$A$_{1g}$) electronic ground state and exhibit typical bonding features of transition metal complexes obeying the 18-electron rule. We also report the isolation and bonding analysis of the charged dinitrogen complexes [M(N$_2$)$_8$]$^+$ (M=Ca, Sr).

[1] Department of Chemistry, Collaborative Innovation Center of Chemistry for Energy Materials, Shanghai Key Laboratory of Molecular Catalysts and Innovative Materials, Fudan University, Shanghai 200433, China. [2] Institute of Advanced Synthesis, School of Chemistry and Molecular Engineering, Jiangsu National Synergetic Innovation Center for Advanced Materials, Nanjing Tech University, Nanjing 211816, China. [3] Fachbereich Chemie, Philipps-Universität Marburg, Hans-Meerwein-Strasse 4, D-35043 Marburg, Germany. [4]These authors contributed equally: Qian Wang, Sudip Pan. Correspondence and requests for materials should be addressed to M.Z. (email: mfzhou@fudan.edu.cn) or to G.F. (email: frenking@chemie.uni-marburg.de)

The chemical conversion of dinitrogen to commercially useful commodities is one of the most important reactions in chemical industry, which has long been the subject of extensive research[1]. Activating the strong triple bond in $N_2$ is a challenge for inventive chemists, who tried in the past various transition metals for binding the molecule as ligand in molecular complexes. Dinitrogen is a poorly binding donor due to its comparatively weak donor–acceptor interactions in transition metal complexes[1,2]. Metal–ligand bonds of $N_2$ are usually much weaker than those of isoelectronic CO. Braunschweig and coworkers recently reported that boron compounds may also be utilized for fixation of dinitrogen[3,4].

Very recently, we reported the isolation and spectroscopic identification of the eight-coordinated carbonyl complexes $M(CO)_8$ of the alkaline earth atoms M=Ca, Sr, Ba possessing cubic ($O_h$) symmetry in a low-temperature matrix[5]. The analysis of the electronic structure showed that the classical main-group metals M bind the CO ligands via donor–acceptor interactions through their $(n)d$ atomic orbitals thus mimicking transition metals. Now we found that the alkaline earth atoms may also bind $N_2$ in octa-coordinated complexes $M(N_2)_8$ (M=Ca, Sr, Ba) whereby the loss of one $N_2$ ligand has surprisingly only a slightly smaller bond dissociation energy than the dissociation of one carbonyl ligand from $M(CO)_8$. Homoleptic dinitrogen complexes of genuine transition metals, such as tetra-coordinated $Ni(N_2)_4$, hexa-coordinated vanadium, and chromium complexes $M(N_2)_6$, have been synthesized and spectroscopically characterized in low-temperature matrices[6–9]. Mass spectrometric and infrared photodissociation spectroscopic investigations in the gas phase revealed that the $Ti^+$, $V^+$, and $Nb^+$ cations form six-coordinated dinitrogen complexes[10–12], while the $Y^+$, $La^+$, and $Ce^+$ cations gave the octa-coordinated dinitrogen complexes[13]. There is only one theoretical report on neutral octa-coordinated dinitrogen complexes by Kovacs, who calculated the lanthanum species $La(N_2)_n$ ($n = 1–8$)[14]. The finding that heavier alkaline earth atoms may bind eight $N_2$ ligands in forming the neutral $M(N_2)_8$ complexes is unprecedented.

## Results

**Experimental studies**. Figure 1 shows the spectra in the terminal N–N stretching frequency region from the experiment using a barium target and a 0.5% $N_2$/Ne sample. The spectra were recorded after (a) 30 min of sample deposition at 4 K, (b) after annealing at 12 K, (c) after 15 min of visible light irradiation, and (d) after 15 min of UV-visible light irradiation. Only two bands at 2118.0 and 2237.6 $cm^{-1}$ were observed. The 2237.6 $cm^{-1}$ band is metal independent and can be assigned to the antisymmetric stretching vibration of the linear $N_4^+$ cation based on the literature data[15]. The 2118.0 $cm^{-1}$ band increases tremendously under UV-visible light irradiation. The weak band at 2141 $cm^{-1}$ is due to trace of CO impurity absorption.

Similar bands centered at 2070.0 and 2057.7 $cm^{-1}$ were observed in the experiments using the strontium and calcium targets. The spectra of strontium and calcium are shown in Supplementary Figs. 1 and 2. Experiments were also performed using the isotopic-substituted $^{15}N_2$ and the $^{14}N_2 + ^{15}N_2$ 1:1 mixture samples. The 2057.7, 2070.0, and 2118.0 $cm^{-1}$ bands are shifted to 1991.0, 2002.7, and 2048.7 $cm^{-1}$, respectively, in the experiments with $^{15}N_2$. The isotopic shifts are appropriate for terminal N–N stretching vibrations. These bands are the only product absorptions in the spectra with high $N_2$ concentrations, suggesting the assignment to the coordinatively saturated $M(N_2)_8$ complexes following the example of $M(CO)_8$[5]. The observation of only one N–N stretching band suggests that these neutral dinitrogen complexes have the highest cubic $O_h$ symmetry. The

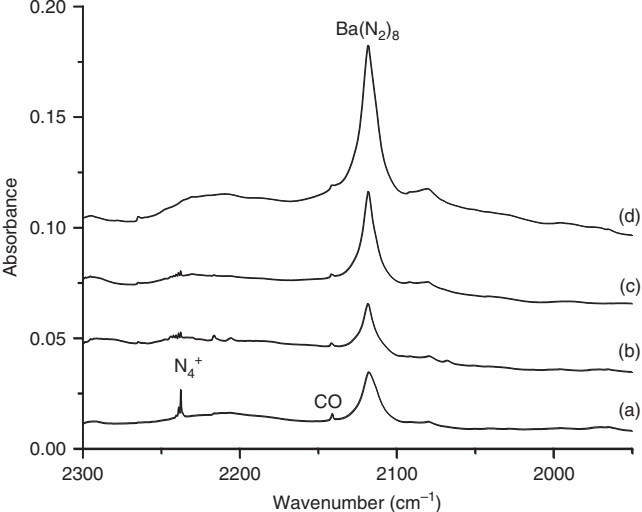

**Fig. 1** Infrared absorption spectra of barium dinitrogen complexes. Infrared absorption spectra in the 2300–1950 $cm^{-1}$ region from co-deposition of laser-evaporated barium atoms with 0.5% $N_2$ in neon. **a** 30 min of sample deposition at 4 K, **b** after annealing at 12 K, **c** after 15 min of visible light irradiation, and **d** after 15 min of UV-visible light irradiation. A weak band at 2141 $cm^{-1}$ is due to trace of CO impurity absorption

isotopic splittings in the experiments with the $^{14}N_2 + ^{15}N_2$ mixed sample cannot be resolved due to band overlap. The spectra of barium (Supplementary Fig. 3) show that two broad bands (2115.4 and 2047.1 $cm^{-1}$) slightly red-shifted from those of pure isotopic counterparts are observed with the $^{14}N_2 + ^{15}N_2$ mixed sample. This mixed isotopic spectral feature is consistent with the cubic structure assignment.

Besides the neutral complexes, the radical cations of the alkaline earth dinitrogen complexes were prepared in the gas phase using a pulsed laser vaporization/supersonic expansion ion source and studied by mass-selected infrared photodissociation spectroscopy in the terminal N–N stretching frequency region (see Supplementary Figs 4–6 for mass spectra)[16]. Each spectrum is composed of a progression of mass peaks that are attributed to the cation complexes $[M(N_2)_n]^+$ (M = Ca, Sr, Ba) with $n$ up to 10. The peaks corresponding to $[Ca(N_2)_8]^+$ and $[Sr(N_2)_8]^+$ are always the most intense peaks at different experimental conditions, suggesting that they are coordination saturated cation complexes. The higher coordinated complexes $[M(N_2)_n]^+$ with $n > 8$ have dinitrogen ligands that are weakly bonded in a second coordination sphere to the $[M(N_2)_8]^+$ core species. Both the $[Ca(N_2)_8]^+$ and $[Sr(N_2)_8]^+$ cation complexes dissociated by elimination of an $N_2$ ligand using a focused IR laser (see Supplementary Fig. 7 for infrared photodissociation spectra). Both spectra feature a broad band centered at 2113 and 2144 $cm^{-1}$, respectively, which are blue-shifted by 56 and 74 $cm^{-1}$ relative to the corresponding neutrals observed in solid neon matrix. The intensities of the $[Ba(N_2)_8]^+$ cation complexes are much lower than those of the Ca and Sr complexes. The $n = 8–11$ complexes are the most intense peaks in the mass spectrum of barium. We are not able to obtain an effective IR spectrum for the $[Ba(N_2)_8]^+$ complex due to its low intensity.

**Theoretical studies**. We performed extensive quantum chemical calculations using density functional theory (DFT) at the M06-2X-D3 and B3LYP-D3 level using various basis set of TZ2P quality in conjunction with relativistic effective core potentials for Sr, and Ba. We also investigated the electronic structure with a modern charge- and energy decomposition analysis to gain

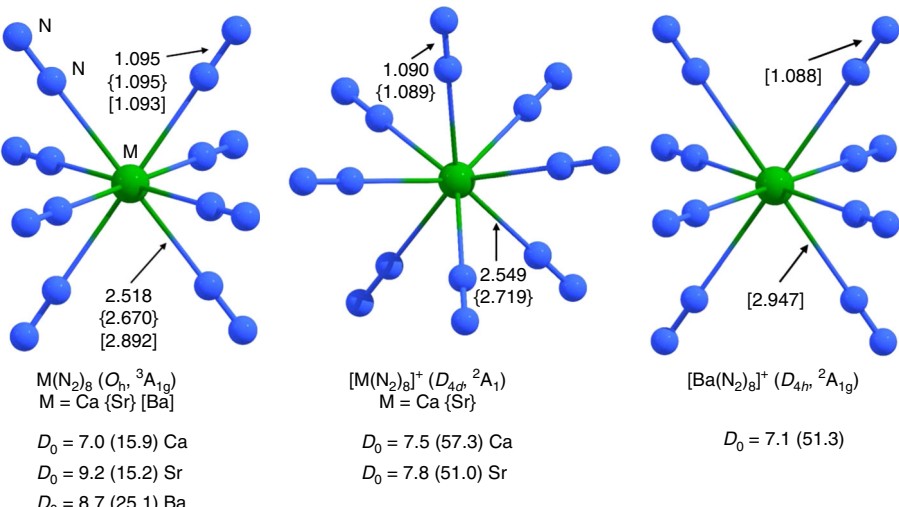

**Fig. 2** Calculated geometries and bond lengths. Calculated equilibrium structures and interatomic distances [Å] of $M(N_2)_8$ and $[M(N_2)_8]^+$ (M = Ca, Sr, Ba) at the M06-2X-D3/def2-TZVPP level. Bond dissociation energies $D_0$ [kcal mol$^{-1}$] for loss of one $N_2$ and (in parentheses) loss of eight $N_2$

insights into the nature of the metal–nitrogen bonds (see Supplementary Methods). Figure 2 shows the energetically lowest lying equilibrium geometries of the $M(N_2)_8$ complexes at the M06-2X-D3/def2-TZVPP level. The atomic coordinates and calculated energies are given in Supplementary Data 1. The molecules have cubic ($O_h$) symmetry and a triplet ($^3A_{1g}$) electronic ground state. The N–N distances in the complexes are slightly longer than in free $N_2$ (1.086 Å). The geometry optimizations of the $M(N_2)_8$ species at the electronic singlet states led to Jahn–Teller distorted structures which have $D_{4d}$ (M = Ca, Sr) or $D_{4h}$ (M = Ba) symmetry (Supplementary Fig. 8). The singlet species are 8.4 kcal mol$^{-1}$ (M = Ca, Sr) and 6.2 kcal mol$^{-1}$ (M = Ba) higher in energy than the triplet complexes. A similar situation was found for the isoelectronic octa-carbonyls $M(CO)_8$, which were calculated at the same level of theory[5].

Figure 2 shows also the bond dissociation energies (BDEs) for loss of one $N_2$ by forming the hepta-coordinated complexes $M(N_2)_7$. The latter species, like the isoelectronic carbonyl complexes[5], have a triplet ground state and a geometry with $C_{3v}$ symmetry, except for $Ca(N_2)_7$ which is slightly distorted from a $C_{3v}$ symmetrical structure (Supplementary Fig. 9). The BDE for the reaction $M(N_2)_8 \rightarrow M(N_2)_7 + N_2$ is between $D_0 = 7.0$ kcal mol$^{-1}$ (M = Ca) and 9.2 kcal mol$^{-1}$ (M = Sr). Surprisingly, the values are only slightly smaller than for the loss of one CO from $M(CO)_8$, which are between $D_0 = 9.1$ kcal mol$^{-1}$ (M = Ca) and 11.5 kcal mol$^{-1}$ (M = Sr) at the same level of theory[5]. However, the total bond strength of eight $N_2$ ligands to the alkaline earth atoms is much lower than for eight CO species (Fig. 2). The BDEs for loss of eight $N_2$ from $M(N_2)_8$ are only between $D_0 = 15.2$ kcal mol$^{-1}$ (M = Sr) and 25.1 kcal mol$^{-1}$ (M = Ba). The corresponding values for loss of eight CO from $M(CO)_8$ are between $D_0 = 58.8$ kcal mol$^{-1}$ (M = Sr) and 63.3 kcal mol$^{-1}$ (M = Ca)[5]. This indicates that the octa-coordinated complexes $M(N_2)_8$ possess a particular stability among the series $M(N_2)_n$, which may be due to a cooperative interaction of all eight ligands. Such cooperative ligand behavior has been reported for mono- and dicarbonyl ion complexes $[M(CO)_2]^+$ with M = Cu, Ag, Au[17].

The BDEs of the eighth $N_2$ are about one-half to one-third of the BDEs for loss of all eight $N_2$. This indicates that the lower coordinated complexes $M(N_2)_n$ ($n < 8$) are much more weakly bonded than $M(N_2)_8$. The calculations predict that the low coordinated adducts $M(N_2)_n$ ($n < 5$) complexes are unstable with respect to the fragments $M + n\,N_2$ in the electronic ground state,

which explains the nearly complete absence of the signals. In the present study, some weak bands are observed in the experiments, which could be due to the appearance of lower coordinated $M(N_2)_n$ complexes with $n = 5$–7, but a definitive assignment is not possible.

The calculated equilibrium geometries and BDEs for the loss of $N_2$ of the radical cations $[M(N_2)_8]^+$ are also given in Fig. 2. The molecules have $D_{4d}$ symmetry for M = Ca and Sr and $D_{4h}$ symmetry for M = Ba; the same symmetry was found for the isoelectronic octa-carbonyl cations $[M(CO)_8]^+$ [5]. The BDEs for loss of one $N_2$ of the cations $[M(N_2)_8]^+$ have similar values as for the neutral complexes $M(N_2)_8$, but the total bond strength for the binding of eight $N_2$ in the cations is much higher than for the neutral species. This explains why signals for nearly all species $[M(N_2)_n]^+$ ($n = 1$–8) are observed in the mass spectra (Supplementary Figs 4–6) while the lower coordinated neutral dinitrogen complexes are not observed. The equilibrium geometries of $[M(N_2)_7]^+$ are shown in Supplementary Fig. 10.

Table 1 shows also the computed N–N stretching frequencies and frequency shifts of the neutral and charged dinitrogen complexes. The calculations give in accordance with experiment a significant red-shift $\Delta v$ of the $M(N_2)_8$ complexes relative to free $N_2$. The theoretical $\Delta v$ values are a bit smaller than the experimental data. We calculated the vibrational frequencies using different functionals and basis sets. The theoretical values at the B3LYP-D3(BJ)/def2-TZVPPD level result in a slightly larger red-shift of the N–N stretching mode than at the M06-2X-D3/def2-TZVPP level, but the corresponding values are still smaller than the experimental results. We think that the discrepancy is likely due to the harmonic approximation of the frequency calculations. Calculations with a different basis set and ECPs for all metal atoms at M06-2X-D3/cc-pCVTZ-pp level gave nearly the same values as the M06-2X-D3/def2-TZVPP calculations (Supplementary Tables 1, 2). The calculated isotope shifts $\Delta v$ ($^{15}N_2$) shown in Table 1 are in excellent agreement with the recorded values. The experiments showed a single N–N-stretching signal for $[Ca(N_2)_8]^+$ and $[Sr(N_2)_8]^+$, which is less red-shifted than for the respective neutral complexes (Table 1). The calculations of $[M(N_2)_8]^+$ suggest two IR-active stretching modes for $N_2$, which are too close to be experimentally observed as separated bands. The calculated wave numbers of the radical cations indicate a smaller red-shift than for the neutral species, which agrees with the observed trend.

**Table 1 Experimental and calculated IR-active N–N stretching frequencies υ of M(N₂)₈ and [M(N₂)₈]⁺ (M = Ca, Sr, Ba) and frequency shifts Δυ (cm⁻¹)**

| | Exptl. | | | | | Calcd.ᵃ | | |
|---|---|---|---|---|---|---|---|---|
| | $\upsilon(^{14}N_2)$ | $\Delta\upsilon^b$ | $\upsilon(^{15}N_2)$ | $\Delta\upsilon^c$ | $\upsilon(^{14}N_2)^d$ | $\Delta\upsilon^b$ | $\upsilon(^{15}N_2)$ | $\Delta\upsilon^c$ |
| Ca(N₂)₈ | 2058 | −272 | 1991 | −67 | 2156 (2124) | −174 (−206) | 2083 (2052) | −73 (−72) |
| Sr(N₂)₈ | 2070 | −260 | 2003 | −63 | 2166 (2123) | −164 (−207) | 2094 (2051) | −72 (−72) |
| Ba(N₂)₈ | 2118 | −212 | 2049 | −69 | 2198 (2135) | −132 (−195) | 2123 (2063) | −75 (−72) |
| [Ca(N₂)₈]⁺ | 2113 | −217 | | | 2233 (2205) | −97 (−125) | | |
| | | | | | 2234 (2206) | −96 (−124) | | |
| [Sr(N₂)₈]⁺ | 2144 | −186 | | | 2252 (2202) | −78 (−128) | | |
| | | | | | 2255 (2203) | −75 (−127) | | |
| [Ba(N₂)₈]⁺ | | | | | 2277 (2217) | −53 (−113) | | |
| | | | | | 2278 (2224) | −52 (−106) | | |

The calculated values are scaled by 0.921 (0.9495). The scaling factor comes from the ratio of the experimental stretching frequency of 2330 cm⁻¹ for N₂ and the calculated value of 2530 cm⁻¹ (2453 cm⁻¹)
ᵃThe calculations were performed at the M06-2X-D3/def2-TZVPP level. The values in parentheses come from B3LYP-D3/def2-TZVPPD calculations
ᵇFrequency shift relative to free N₂. The experimental value for N₂ is 2330 cm⁻¹ and the calculated value is 2530 cm⁻¹
ᶜIsotope frequency shift
ᵈFrequency of the IR-active t₁ᵤ mode

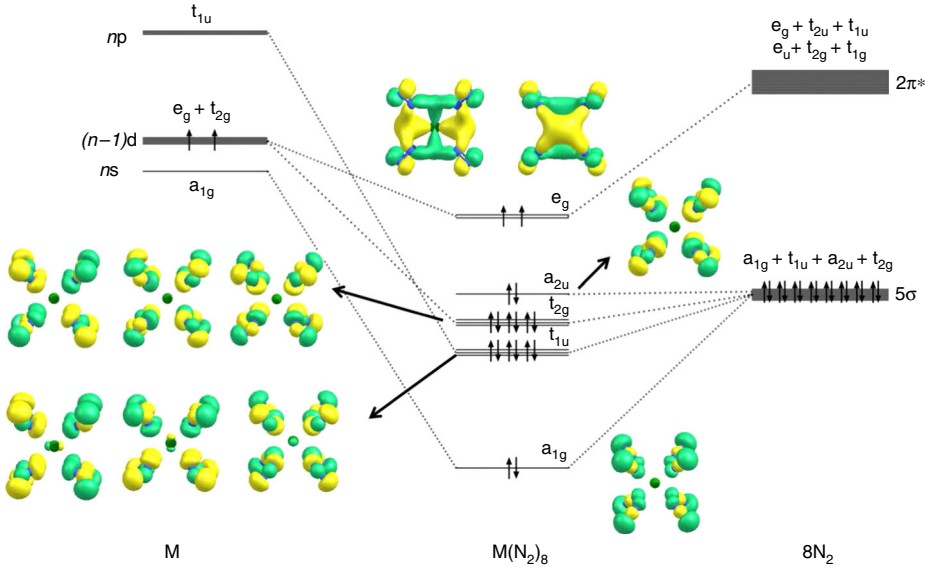

**Fig. 3** Orbital correlation diagram. Orbital correlation diagram of the spd valence orbitals of an atom M with the configuration $(n-1)d^2(n)s^0(n)p^0$ in an octa-coordinated cubic ($O_h$) field of eight N₂ ligands and occupied valence orbitals of Ca(N₂)₈. Only the occupied valence orbitals that are relevant for the Ca–N₂ interactions are shown

Figure 3 shows the orbital correlation diagram M with a spd valence shell and the electron configuration $(n-1)d^2ns^0np^0$ in the cubic ($O_h$) field of eight N₂ ligands. It explains the paradoxical situation that only 16 valence electrons of the 18-electron species M(N₂)₈ (M=Ca, Sr, Ba) are used to fill the valence shell of the metals. The HOMO (highest occupied molecular orbital) is degenerate; the occupation by two electrons with the same spin gives a stable electron configuration like in the $X^3\Sigma_g^-$ electronic ground state of O₂. Filling the $e_g$ HOMO with two more electrons gives the 20 valence electron system ML₈, which fulfills the 18-electron rule. Examples of the latter system were recently reported by us in an experimental/theoretical study of [TM(CO)₈]⁻ (TM=Sc, Y, La)[18].

The orbital interaction of a metal with a spd valence shell and eight ligands L in a cubic ($O_h$) field gives valence orbitals that have $a_{1g}$, $t_{1u}$, $t_{2g}$, $a_{2u}$, and $e_g$ symmetry, which can be associated with specific valence AOs of the metal (Fig. 3). The strength of the

pairwise orbital interactions may be estimated with the EDA-NOCV (Energy Decomposition Analysis in combination with Natural Orbitals for Chemical Valence) method. Details of the method and further examples have been described in the literature[19,20]. Table 2 gives the numerical results of the calculations of M(N₂)₈ (M=Ca, Sr, Ba) using the fragments M in the electronic triplet state with $(n-1)d^2ns^0np^0$ electron configuration and (N₂)₈ in the singlet state.

The data in Table 2 show that the intrinsic interaction energies $\Delta E_{int}$ of M(N₂)₈ between the metal atoms in the electronic reference state and the dinitrogen ligand cage are rather large. They are much larger than the bond dissociation energies, which are only between 15.2 kcal mol⁻¹ for M=Sr and 25.1 kcal mol⁻¹ for M=Ba. The BDE values give the energy difference between the complex and the fragments at the equilibrium geometries in the electronic ground state, whereas the interaction energies $\Delta E_{int}$ refer to the energy difference between the complex and the fragments in

**Table 2 EDA-NOCV results for triplet $M(N_2)_8$ (M=Ca, Sr, Ba) complexes at the M06-2X/TZ2P//M06-2X-D3/def2-TZVPP level**

| Energy terms | Orbital interactions | Ca (T) + $(N_2)_8$ (S) ($O_h$) | Sr (T) + $(N_2)_8$ (S) ($O_h$) | Ba (T) + $(N_2)_8$ (S) ($O_h$) |
|---|---|---|---|---|
| $\Delta E_{int}$ | | −190.7 | −175.2 | −104.0 |
| $\Delta E_{hybrid}$ | | 48.9 | 51.4 | 35.1 |
| $\Delta E_{Pauli}$ | | 25.8 | 31.3 | 37.0 |
| $\Delta E_{elstat}$[a] | | −49.2 (18.5%) | −45.2 (17.5%) | −54.7 (31.1%) |
| $\Delta E_{orb}$[a] | | −216.2 (81.5%) | −212.5 (82.5%) | −121.4 (68.9%) |
| $\Delta E_{orb(1)}$[b] | $[M(d)]{\rightarrow}(N_2)_8$ π backdonation | −184.3 (85.2%) | −180.6 (85.0%) | −85.0 (70.0%) |
| $\Delta E_{orb(2)}$[b] | $[M(d)]{\leftarrow}(N_2)_8$ σ donation | −18.0 (8.3%) | −17.4 (8.2%) | −18.0 (14.8%) |
| $\Delta E_{orb(3)}$[b] | $[M(s)]{\leftarrow}(N_2)_8$ σ donation | −3.5 (1.6%) | −3.9 (1.8%) | −3.7 (3.0%) |
| $\Delta E_{orb(4)}$[b] | $[M(p)]{\leftarrow}(N_2)_8$ σ donation | −2.4 (1.1%) | −2.7 (1.3%) | −4.2 (3.5%) |
| $\Delta E_{orb(5)}$[b] | $(N_2)_8$ polarization | −0.8 (0.4%) | −1.1 (0.5%) | −1.9 (1.6%) |
| $\Delta E_{orb(rest)}$ | | −7.2 (3.4%) | −6.8 (3.2%) | −8.6 (7.1%) |

The interacting fragments are the metal atom M in the triplet excited state with a $(n)s^0(n-1)d^2$ valence electronic configuration and $(N_2)_8$ in the singlet state. Energy values are given in kcal mol$^{-1}$
[a]The values in parentheses give the percentage contribution to the total attractive interactions $\Delta E_{elstat} + \Delta E_{orb}$
[b]The values in parentheses give the percentage contribution to the total orbital interactions $\Delta E_{orb}$

---

**Table 3 Experimental excitation energies of the alkaline atoms M and ions $M^+$ from the electronic ground state to the reference state in the complexes $M(N_2)_8$ and $[M(N_2)_8]^+$ (M = Ca, Sr, Ba)**

| | Excitation[a] | Ca | Sr | Ba |
|---|---|---|---|---|
| Neutral atom | $(n)s^2(n-1)d^0 \rightarrow (n)s^0(n-1)d^2$ | 124.2[b] | 127.2[c] | 59.8[d] |
| Cation | $(n)s^1(n-1)d^0 \rightarrow (n)s^0(n-1)d^1$ | 39.0[b] | 41.6[e] | 13.9[d] |

Values are given in kcal mol$^{-1}$
[a]The electronic ground states are $^1S$ for the neutral atoms M and $^2S$ for the ions $M^+$. The lowest lying excited reference states of the neutral atoms are $^3F$ for Ca and Ba and $^3P$ for Sr. The lowest lying excited reference state of the cations $M^+$ is $^2D$. The cited values refer to the lowest J level
[b]ref. 29
[c]ref. 30
[d]ref. 31
[e]ref. 32

the frozen geometry and the electronic reference state. The electronic reference state of the metal atoms is the excited triplet state with the electron configuration $(n)s^0(n-1)d^2$ and the electronic ground state is a singlet with the configuration $(n)s^2(n-1)d^0$. The excitation energies $(n)s^2(n-1)d^0 \rightarrow (n)s^0(n-1)d^2$ of the alkaline earth atoms M are rather large for M=Ca, Sr and notably smaller for M=Ba (Table 3). This explains why $Ba(N_2)_8$ has a higher BDE than $Ca(N_2)_8$ and $Sr(N_2)_8$ although the interaction energy $\Delta E_{int}$ of the former species is significantly weaker than those of the lighter homologs.

The attractive metal–ligand interactions $\Delta E_{int}$ of $M(N_2)_8$ come mainly from the covalent (orbital) term $\Delta E_{orb}$. The breakdown of the latter term into pairwise orbital interactions $\Delta E_{orb(1)} - \Delta E_{orb(5)}$ shows that the covalent bonding is dominated by the $[M(d)]{\rightarrow}(N_2)_8$ π backdonation $\Delta E_{orb(1)}$, which contributes 70–85% to the total orbital interactions. The rather diffuse $(n-1)d$ orbitals give the alkaline earth elements Ca–Ba an unusual reactivity with a remarkable donor ability. This was previously noted in our study of the carbonyl cation complexes $[Ba(CO)]^+$ where the metal cation $Ba^+$ is a donor for the neutral CO ligand[21]. We are not aware of any other metal cations that donate electronic charge to a neutral acceptor.

Figure 3 shows also the occupied MOs of $Ca(N_2)_8$. Visual inspection of the orbital shapes suggests that the contribution of the d-AOs of Ca in the $e_g$ HOMO is much larger than those of the calcium orbitals in the other MOs. The dominant role of the $d(\pi)$-AOs of Ca comes to the fore by the deformation densities $\Delta\rho_{(1)-(5)}$, which are associated with the pairwise orbital

interactions $\Delta E_{orb(1)-(5)}$ (Fig. 4). Note that the isosurface values for $\Delta\rho_{(2)-(5)}$ are much smaller than for $\Delta\rho_{(1)}$, because otherwise the small contributions of the metal AOs would not be visible. The relative size of the charge transfer is given by the eigenvalues of the deformation densities $|\upsilon_n|$. It becomes obvious that the largest charge transfer occurs for the $[M(d)]{\rightarrow}(N_2)_8$ π backdonation $|\upsilon_1|$. The color code for the charge transfer is red→blue. The shape of the deformation densities $\Delta\rho_{(2)-(5)}$ indicates that the orbital interactions $\Delta E_{orb(2)-(5)}$ include charge transfer (polarization) within the $N_2$ ligands with $\Delta E_{orb(5)}$ exclusively coming from polarization. The deformation densities $\Delta\rho_{(1)-(5)}$ of the heavier complexes $M(N_2)_8$ (M = Sr, Ba) are shown in Supplementary Figs. 11 and 12.

We also analyzed the nature of the metal–ligand interactions in the cations $[M(N_2)_8]^+$ using $M^+$ in the electronic reference state $^2D$ with the electron configuration $(n)s^0(n-1)d^1$ and the ligand cage $(N_2)_8$ as interacting fragments. The numerical results are shown in Table 4. The calculated interaction energies $\Delta E_{int}$ of the cations are much smaller than in the neutral molecules (Table 2). This could be expected, because the dominant orbital interaction in $M(N_2)_8$ comes from the $[M(d)]{\rightarrow}(N_2)_8$ π backdonation of the doubly occupied $e_g$ MO (Fig. 3). There is only one valence electron in $M^+$ which is also less inclined to π backdonation due to the positive charge of the metal. Table 4 shows that the contribution of the $[M^+(d)]{\rightarrow}(N_2)_8$ π backdonation is much weaker than the related term in the neutral molecules (Table 2). The lower symmetry of $[M(N_2)_8]^+$ ($D_{4d}$ or $D_{4h}$) leads to a splitting of the orbital terms. The sum of the two σ donations into the metal d-AOs $[M(d)]^+{\leftarrow}(N_2)_8$ is slightly higher than in the neutral complexes, but it does not compensate for the weaker π backdonation. The cations $[M(N_2)_8]^+$ have a significantly higher BDE than the neutral complexes (Fig. 2), because the excitation energy into the $^2D$ electronic reference state of $M^+$ is much less than the promotion to the electronic reference state of the neutral atoms (Table 3). It is interesting to note that the $^2D$ state with the configuration $(n)s^0(n-1)d^1$ is the first excited state of the metal ions $M^+$. The deformation densities $\Delta\rho_{(1)-(7)}$, which are associated with the pairwise orbital interactions $\Delta E_{orb(1)-(7)}$ of the cations $[M(N_2)_8]^+$ show the expected features of the charge flow. They are shown in Supplementary Figs 13–15.

One referee noted reservation against the use of the 18 valence electron rule for the systems, saying that the coordination number 8 is very favorable for the heavier alkaline earth atoms. We want to point out that this refers to compounds in the solid state, where the alkaline earth atoms are found as isolated $M^{2+}$ cations, which are stabilized by the ligand cage of the solid. Isolated molecules do not possess ionic bonds but polar covalent bonds. Diatomic CaO

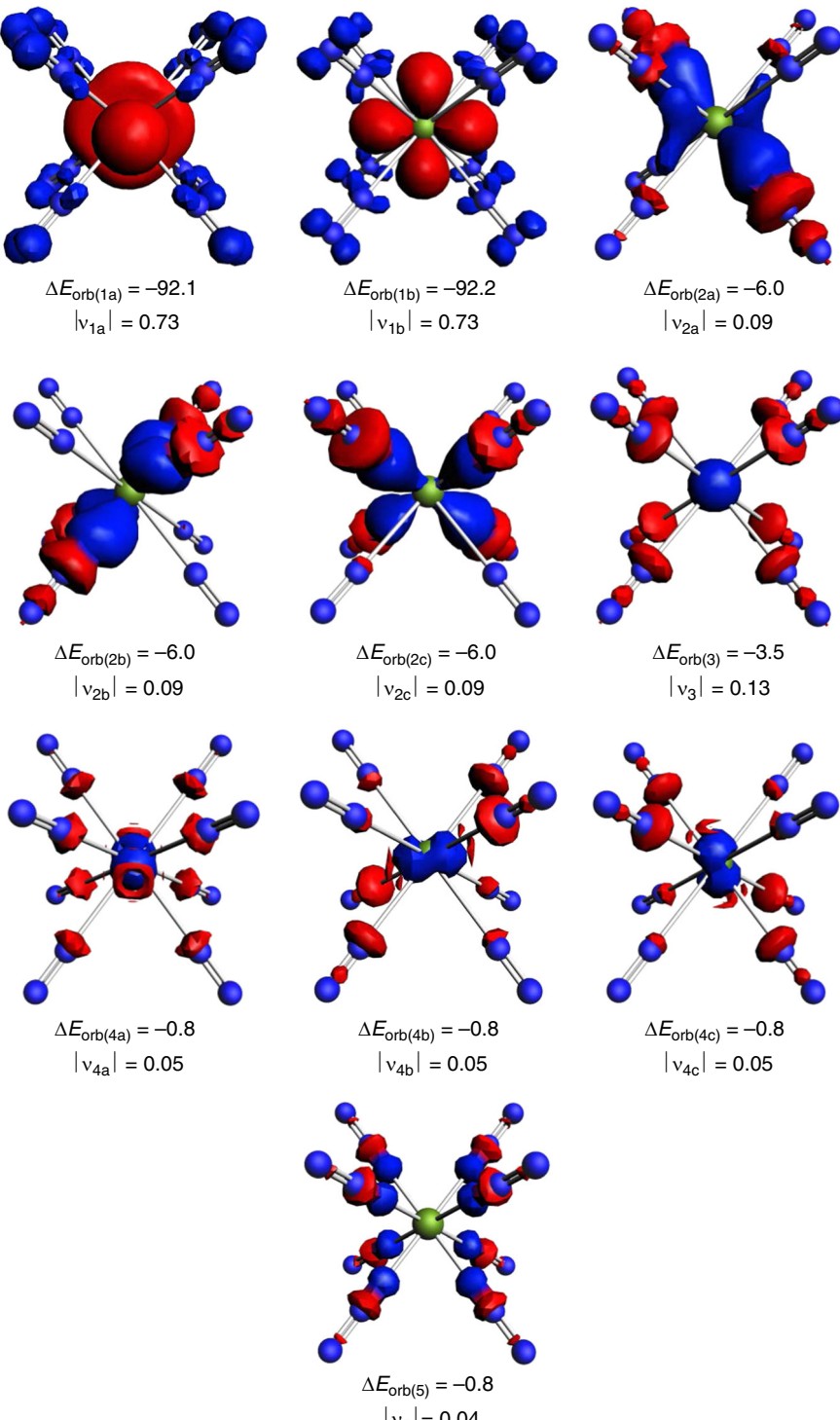

**Fig. 4** Shape of deformation densities. Shape of the deformation densities $\Delta\rho_{(1)-(5)}$, which are associated with the orbital interactions $\Delta E_{orb(1)-(5)}$ in $Ca(N_2)_8$ (Table 2) and eigenvalues $|\nu_n|$ of the charge flow. The isosurface values are 0.002 for $\Delta\rho_{(1)}$ and 0.0006 for $\Delta\rho_{(2)-(5)}$. The color code of the charge flow is red → blue

has a bond length of 1.821 Å[22], whereas solid CaO has a Ca–O distance of 2.42 Å[23]. The latter value agrees with an ionic bond, whereas the former data comes from a polar covalent double bond. The atomic partial charges at the metal atom calculated by the NBO 6.0 method[24] are 1.25 (Ca), 1.24 (Sr), and 1.05 (Ba), which also indicate polar covalent bonds rather ionic bonds. The electrostatic attraction in ionic solids comes from the mutual attraction of the separated ions, whereas the electrostatic

component in the EDA-NOCV calculations comes from inter-penetrating charges. We want to emphasize that the data in Table 2 consider the metal–ligand interactions in terms of M←L σ donation and M→L π backdonation between the fragments in the electronic reference state prior to bond formation, which agrees with the Dewar–Chatt–Duncanson (DCD) model[25–27]. We think that the bonding situation in the isolated molecules is best described in terms of polar covalent bonds due to dative

**Table 4 EDA-NOCV results for doublet [M(N$_2$)$_8$]$^+$ (M = Ca, Sr, Ba) complexes at the M06-2X/TZ2P-ZORA//M06-2X-D3/def2-TZVPP level**

| Energy | Orbital interactions | Ca$^+$ (D) + (N$_2$)$_8$ (S) ($D_{4d}$) | Sr$^+$ (D) + (N$_2$)$_8$ (S) ($D_{4d}$) | Ba$^+$ (D) + (N$_2$)$_8$ (S) ($D_{4h}$) |
|---|---|---|---|---|
| $\Delta E_{int}$ | | −106.7 | −100.3 | −74.3 |
| $\Delta E_{hybrid}$ | | 27.0 | 27.0 | 17.2 |
| $\Delta E_{Pauli}$ | | 38.3 | 38.6 | 48.6 |
| $\Delta E_{elstat}$[a] | | −48.3 (28.1%) | −45.7 (27.5%) | −49.3 (35.2%) |
| $\Delta E_{orb}$[a] | | −123.7 (71.9%) | −120.2 (72.5%) | −90.9 (64.8%) |
| $\Delta E_{orb(1)}$[b] | [M($d$)]$^+$→(N$_2$)$_8$ π backdonation | −49.9 (40.3%) | −55.7 (46.3%) | −36.0 (39.6%) |
| $\Delta E_{orb(2)}$[b] | [M($d$)]$^+$←(N$_2$)$_8$ σ donation | −17.4 (14.1%) | −16.2 (13.5%) | −14.6 (16.1%) |
| $\Delta E_{orb(3)}$[b] | [M($d$)]$^+$←(N$_2$)$_8$ σ donation | −13.6 (11.0%) | −11.2 (9.3%) | −6.2 (6.8%) |
| $\Delta E_{orb(4)}$[b] | [M($s$)]$^+$←(N$_2$)$_8$ σ donation | −6.4 (5.2%) | −6.0 (5.0%) | −4.4 (4.8%) |
| $\Delta E_{orb(5)}$[b] | [M($p$)]$^+$←(N$_2$)$_8$ σ donation | −7.0 (5.7%) | −5.4 (4.5%) | −4.4 (4.8%) |
| $\Delta E_{orb(6)}$[b] | [M($p$)]$^+$←(N$_2$)$_8$ σ donation | −3.0 (2.4%) | −3.5 (2.9%) | −2.2 (2.4%) |
| $\Delta E_{orb(7)}$[b] | (N$_2$)$_8$ polarization | -c- | -c- | −2.5 (2.8%) |
| $\Delta E_{orb(rest)}$ | | −26.4 (21.3%) | −22.2 (18.5%) | −20.6 (22.7%) |

The interacting fragments are the metal cation M$^+$ in the doublet excited state with a ($n$)s$^0$($n$−1)d$^1$ valence electronic configuration and (N$_2$)$_8$ in the singlet state. Energy values are given in kcal mol$^{-1}$
[a]The values in parentheses give the percentage contribution to the total attractive interactions $\Delta E_{elstat}$ + $\Delta E_{orb}$
[b]The values in parentheses give the percentage contribution to the total orbital interactions $\Delta E_{orb}$
[c]In $D_{4d}$ field, The (N$_2$)$_8$ polarization does not correlate with a specific orbital symmetry; it is part of $\Delta E_{orb(3)}$, which comprises mainly [M($d$)]$^+$←(N$_2$)$_8$ σ donation

interactions that can be described with the DCD model and which agree with the 18-electron rule.

In summary, we report the first octa-coordinated heavy alkaline earth metal–dinitrogen complexes M(N$_2$)$_8$ (M = Ca, Sr, Ba) that feature strong red-shift in the N–N stretching frequencies and also obey the 18-electron rule. We also report the molecular ions [M(N$_2$)$_8$]$^+$ and provide a detailed analysis of the bonding situation in the complexes. The results suggest new perspectives for the topic of dinitrogen activation.

## Methods

The alkaline earth metal–dinitrogen complexes were prepared via the reactions of pulsed laser-evaporated alkaline earth metal atoms and dinitrogen molecules in solid neon. The product species were detected by infrared absorption spectroscopy employing a Bruker Vertex 80 V spectrometer at 0.5 cm$^{-1}$ resolution using a liquid-nitrogen cooled mercury cadmium telluride (MCT) detector. The experiments were carried out with a wide range of dinitrogen concentrations (from 0.02 to 2% relative to Ne on the basis of volume). No obvious product absorptions were observed in the experiments with very low N$_2$ concentrations (<0.05%). Further details have been described before[28].

## Data availability

All the studied data are presented in the manuscript and supplementary materials. Additional raw data that support the findings of this study are available from thecorresponding authors upon reasonable request.

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

## Acknowledgements

The experimental work was supported by the National Natural Science Foundation of China (grant numbers 21688102 and 21433005). L.Z. and G.F. acknowledge financial support from Nanjing Tech University (grant numbers 39837123 and 39837132) and SICAM Fellowship from Jiangsu National Synergetic Innovation Center for Advanced Materials, Natural Science Foundation of Jiangsu Province for Youth (grant number BK20170964), National Natural Science Foundation of China (grant number 21703099). S.P. thanks Nanjing Tech University for a postdoctoral fellowship and the high performance center of Nanjing Tech University for the computational resources. G.F. and S.P. are grateful to the Deutsche Forschungsgemeinschaft for financial support.

## Author contributions

S.P. carried out the quantum chemical calculations. L.Z. and G.F. analyzed the theoretical data and wrote the theoretical part of the paper. Q.W., G.D., and G.W. did the matrix isolation experiments, S.L. and J.J. did the gas phase infrared photodissociation experiments. M.Z. analyzed the experimental data and wrote the experimental part of the paper. M.Z. and G.F. supervised the experimental and theoretical work. Q.W. and S.P. contributed equally to this work.

## Additional information

**Competing interests:** The authors declare no competing interests.

