## [Transparent Peer Review File · Nature Communications]

Reviewers' comments:

Reviewer #1 (Remarks to the Author):

The manuscript of Zhou, Frenking and coworkers details the gas- and matrix-phase generation of $M(N_2)_8$ complexes (M = alkaline earth metals). The authors prepare a range of $M(N_2)_8$ species by reactions of metal atoms with N_2 in solid neon. The corresponding radical cations were prepared in the gas phase and detected by mass-selected infrared photodissociation spectroscopy. Although the complexes are only transient and will not be stable under normal laboratory conditions, the results are still quite surprising and interesting. The detection and theoretical studies appear to be reliable, and the manuscript is well constructed. For these reasons I believe the work is suitable for publication in Nature Communications. I only have a few very minor corrections that the authors should make before publication.

- The new Science paper on N_2 catenation from Braunschweig should be added in addition to Ref 3.

- Line 140 - There is a reference that is not defined here.

Reviewer #2 (Remarks to the Author):

This is a very interesting manuscript that will clearly arise a lot of attention and thus can be published with Nature Comm.

However, for me the 18 VE claim is rather far fetched and several inconsistencies in the description force me to select major revision. Please address the points raised below in a revised version, augment and comment.

Points to be addressed in a revised version:

(i) As for the mass spectra:

The intensities especially for the $Ba(N_2)_x^+$ case in Extended Data Fig. 6. do not stop at $x = 8$, but rather continue with significant intensities up to $x = 12$, with $x = 11$ apparently having intensities higher than that of the investigated $Ba(N_2)_8^+$. Why did the authors not calculate those and discuss their properties? The authors mention the 18 VE rule as being important. However, the above numbers for x in no case have any meaning in terms of the 18 VE rule.

(ii) The authors state "The intensities of the $[Ba(N_2)_8]^+$ cation complexes are much lower than those of the Ca and Sr complexes. The $n = 8-11$ complexes are the most intense peaks in the mass spectrum of barium. We are not able to obtain an effective IR spectrum for the $[Ba(N_2)_8]^+$ complex due to its low intensity. "

However, if one looks at the intensity scale bars in the extended Data figures 4 to 6, actually the intensities of the Ba-complexes are the highest of all ($Ca < 40$, $Sr < 30$, Ba approx. 300!, each for the octa- N_2 -complexes). Please account for this.

(iii) Fig. 1: I presume the small signal at 2143 cm^{-1} is attributed to free CO which comes as an impurity of the N_2 gas...? Please label the band and add a sentence to the figure caption.

(iv) Your statement "Filling the eg HOMO with two more electrons gives the 20 valence electron system ML_8 , which fulfills the 18-electron rule."

I do not understand. How can a 20 VE complex fulfil the 18 VE rule. From the MO diagram it

becomes clear, that 16 from the 18 VE in $M(N_2)_8$ are used to form the ligand group orbitals of a cubic array of 8 dinitrogen ligands, thus a linear combination of eight times the HOMO of N_2 in cubic arrangement. This fully complies with your NOCV analysis. However, the only orbital truly relevant to M- N_2 bonding is the e_g orbital including two single electrons. It is fully M- N_2 bonding, but N-N antibonding. Thus, the optimum number for a most stable $M(N_2)_8(y)$ should be at $y = -2$, in which this e_g orbital would be fully occupied.

I am aware that a dianion is very difficult to realize in the MS, but wouldn't the monoanion with $y = -1$ not be a very relevant case that should be studied by MS and quantum chemistry...?

In this respect the $y = +1$ case seems to be less relevant. However, its total binding energy is much higher than that of the neutral case with $y = 0$. This suggests that also the orbital energies must react favorably to the positive ionization, as else it would be expected that the reduction from two bonding electrons in the e_g orbital to one electron in the monocation would also lead to a large reduction of bond energy. Please use simple orbital energy arguments, augment and explain.

In any case, I have a hard time to see the special stability of the 18 VE case. Please reconsider and explain.

Reviewer #3 (Remarks to the Author):

This paper presents a combined experimental and computational study of a remarkable series of homoleptic $M(N_2)_8$ complexes of the alkaline earths. The experimental work is high quality, and the computational analysis is insightful. In isolation, then, I would have no hesitation in recommending publication. My only real concern is that the paper is strikingly similar to the authors' recent Science article on $M(CO)_8$ analogues: the first three figures are almost identical in format, as are the two tables, and the arguments flow along the same lines. The fact that it is N_2 brings in a different context (nitrogen fixation) but from a fundamental scientific perspective the similarities are striking. On balance, my opinion is that the very high novelty of the Science paper raises the profile of this follow-up, and so I would be happy to see it published in Nature Comm.

I have a few small points below that the authors might consider:

1) Is the relatively low stability of $M(N_2)_x$ where $x < 8$ connected with the switch from $4s^2$ to $3d^2$ configuration (which, presumably, allows for much closer approach of the N_2 ligands). It would be useful to know whether all of the complexes studied have a d^2 configuration: if not, where is the transition point?

2) Are the orbital interactions (and deformation densities) symmetry based (i.e. is ΔE_{orb1} strictly synonymous with e_g , ΔE_{orb2} with t_{2g} etc)?

Reply to the referee comments and list of changes.

We are grateful to the referee comments for the positive remarks and for the constructive suggestions, which helped to improve the paper. Below is our reply to the comments and the list of changes.

Reviewer #1 (Remarks to the Author):

The manuscript of Zhou, Frenking and coworkers details the gas- and matrix-phase generation of $M(N_2)_8$ complexes (M = alkaline earth metals). The authors prepare a range of $M(N_2)_8$ species by reactions of metal atoms with N_2 in solid neon. The corresponding radical cations were prepared in the gas phase and detected by mass-selected infrared photodissociation spectroscopy. Although the complexes are only transient and will not be stable under normal laboratory conditions, the results are still quite surprising and interesting. The detection and theoretical studies appear to be reliable, and the manuscript is well constructed. For these reasons I believe the work is suitable for publication in Nature Communications. I only have a few very minor corrections that the authors should make before publication.

- The new Science paper on N_2 catenation from Braunschweig should be added in addition to Ref 3.

Reply: The new Science paper by Braunschweig has been added as new ref. 4

- Line 140 - There is a reference that is not defined here.

Reply:
This has been corrected,

Reviewer #2 (Remarks to the Author):

This is a very interesting manuscript that will clearly arise a lot of attention and thus can be published with Nature Comm.

However, for me the 18 VE claim is rather far fetched and several inconsistencies in the description force me to select major revision. Please address the points raised below in a revised version, augment and comment.

Points to be addressed in a revised version:

(i) As for the mass spectra:

The intensities especially for the $\text{Ba}(\text{N}_2)_x^+$ case in Extended Data Fig. 6. do not stop at $x = 8$, but rather continue with significant intensities up to $x = 12$, with $x = 11$ apparently having intensities higher than that of the investigated $\text{Ba}(\text{N}_2)_8^+$. Why did the authors not calculate those and discuss their properties? The authors mention the 18 VE rule as being important. However, the above numbers for x in no case have any meaning in terms of the 18 VE rule.

Reply: The cation complexes are prepared via a pulsed laser evaporation-supersonic expansion ion source. The $\text{M}(\text{N}_2)_x^+$ complexes up to $x=12$ or more are detected in the mass spectra. The N_2 ligands in these highly coordinated complexes cannot all be directly coordinated to the central metal ion. These complexes have “external” N_2 ligands not directly bound to the central metal ion, but coordinated to the core ion via weaker electrostatic forces. The formation of these “solvated” complexes is possible only because of the cold supersonic expansion conditions. Our calculations predict that the $x=8$ complex is the coordination saturated ions, and the $x>8$ ions are weakly bound complexes involving a $\text{M}(\text{N}_2)_8^+$ core ion where the further N_2 ligands are weakly tagged in the second ligand sphere. A sentence has been added on page 3/4 saying “The higher coordinated complexes $[\text{M}(\text{N}_2)_n]^+$ with $n > 8$ have dinitrogen ligands that are weakly bonded in a second coordination sphere to the $[\text{M}(\text{N}_2)_8]^+$ core species.”

(ii) The authors state "The intensities of the $[\text{Ba}(\text{N}_2)_8]^+$ cation complexes are much lower than those of the Ca and Sr complexes. The $n = 8-11$ complexes are the most intense peaks in the mass spectrum of barium. We are not able to obtain an effective IR spectrum for the $[\text{Ba}(\text{N}_2)_8]^+$ complex due to its low intensity. "

However, if one looks at the intensity scale bars in the extended Data figures 4 to 6, actually the intensities of the Ba-complexes are the highest of all (Ca < 40, Sr < 30, Ba approx. 300!, each for the octa- N_2 -complexes). Please account for this.

Reply: We are sorry that we did not present the experimental results clear enough, which might have caused some misunderstanding. The intensity scale bars in Figures S4-S6 were plotted in arbitrarily unit and are not comparable. The figures are now replotted with comparable intensity scale bars.

(iii) Fig. 1: I presume the small signal at 2143 cm^{-1} is attributed to free CO which comes as an impurity of the N_2 gas...? Please label the band and add a sentence to the figure caption.

Reply: The CO absorption is observed in all matrix isolation experiments, which comes from trace of impurity in the samples. The band at 2143 cm^{-1} is now labeled in Figure 1 and a sentence is added in the caption.

(iv) Your statement "Filling the eg HOMO with two more electrons gives the 20 valence electron system ML_8 , which fulfills the 18-electron rule."

I do not understand. How can a 20 VE complex fulfill the 18 VE rule. From the MO diagram it becomes clear, that 16 from the 18 VE in $M(N_2)_8$ are used to form the ligand group orbitals of a cubic array of 8 dinitrogen ligands, thus a linear combination of eight times the HOMO of N_2 in cubic arrangement. This fully complies with your NOCV analysis. However, the only orbital truly relevant to M- N_2 bonding is the eg orbital including two single electrons. It is fully M- N_2 bonding, but N-N antibonding. Thus, the optimum number for a most stable $M(N_2)_8(y)$ should be at $y = -2$, in which this eg orbital would be fully occupied.

Reply: The 18-electron rule states that the (n)s(n)p(n-1)d valence orbitals of a transition metal in complexes ML_n are all filled by the ligands L. *This applies regardless of whether the metal-ligand interactions are weak or strong!* However, the symmetry of the MOs could prevent that the occupied ligand orbitals donate electronic charge into the valence orbitals of the transition metal. This holds for the a_{2u} MO of cubic (O_h) $M(N_2)_8$ (see Fig. 3). But since the HOMO is a degenerate e_g MO, where both components are occupied by one electron, all valence orbitals of the metal are at least partially occupied. Thus, 16 of the 18 valence electrons of $M(N_2)_8$ are sufficient to fill the valence shell of the metal. A related case is found in square planar complexes. This is described and discussed in the book by Albright, Whangbo and Burdett "Orbital Interactions in Chemistry" and in older papers by Roald Hoffmann. If the e_g HOMO of cubic ML_8 would be completely occupied, the molecule would have 20 valence electrons of which only 18 occupy the metal AOs; the 18-electron rule is fulfilled. What counts is the symmetry, NOT the strength of the metal-ligand interactions.

I am aware that a dianion is very difficult to realize in the MS, but wouldn't the monoanion with $y = -1$ not be a very relevant case that should be studied by MS and quantum chemistry...?

Reply: We did experiments trying to prepare the $M(N_2)_8^-$ monoanion complexes in the gas phase. Unfortunately, we were not able to observe any $M(N_2)_n^-$ anion species in the mass spectra, implying that the $M(N_2)_n^-$ species cannot be formed in our experimental conditions.

In this respect the $y = +1$ case seems to be less relevant. However, its total binding energy is much higher than that of the neutral case with $y = 0$. This suggests that also the orbital energies must react favorably to the positive ionization, as else it would be expected that the reduction from two bonding electrons in the eg orbital to one electron in the monocation

would also lead to a large reduction of bond energy. Please use simple orbital energy arguments, augment and explain.

Reply: Good point!! We are very grateful for the suggestion. We did not pay attention to the finding that the monocations $M(N_2)_8^+$ has a larger total BDE than the neutral systems $M(N_2)_8$ although the most important e_g MO of the latter complexes for metal-ligand binding is now only singly occupied. We carried out EDA-NOCV calculations of the radical cations and found that the intrinsic $M^+-(N_2)_8$ binding interactions are WEAKER than in the neutral species. The reason why the BDE of the cations is higher than in the neutral systems comes from the much larger smaller excitation energy of the metal ions M^+ from the $(n)s^1$ (2S) ground state to the $(n)d^1$ (2D) reference state in the complexes compared with the excitation energy of the neutral metal M from the $(n)s^2$ (1S) ground state to the $(n)d^2$ (2F) reference state. We present the data in the new tables 3 and 4 and we discuss the findings in the new yellow marked text on pages 7 and 8. This is a significant extension of the work, which is acknowledged in the manuscript.

In any case, I have a hard time to see the special stability of the 18 VE case. Please reconsider and explain.

Reply: See our reply above. The unsaturated complexes shall be studied in a forthcoming work. See also our reply to comment 1 of reviewer 3 below.

Reviewer #3 (Remarks to the Author):

This paper presents a combined experimental and computational study of a remarkable series of homoleptic $M(N_2)_8$ complexes of the alkaline earths. The experimental work is high quality, and the computational analysis is insightful. In isolation, then, I would have no hesitation in recommending publication. My only real concern is that the paper is strikingly similar to the authors' recent Science article on $M(CO)_8$ analogues: the first three figures are almost identical in format, as are the two tables, and the arguments flow along the same lines. The fact that it is N_2 brings in a different context (nitrogen fixation) but from a fundamental scientific perspective the similarities are striking. On balance, my opinion is that the very high novelty of the Science paper raises the profile of this follow-up, and so I would be happy to see it published in Nature Comm.

I have a few small points below that the authors might consider:

1) Is the relatively low stability of $M(N_2)_x$ where $x < 8$ connected with the switch from $4s^2$ to $3d^2$ configuration (which, presumably, allows for much closer approach of the N_2 ligands). It would be useful to know whether all of the complexes studied have a d^2 configuration: if not, where is the transition point?

Reply: The question of the referee addresses a very important topic, which is, however, beyond the scope of this work. We think that the excitation energy of the alkaline atoms to excited states where the d-AOs are occupied is crucially connected to the structure and stability of the complexes. We believe that the rather low excitation energy of the metal cation M^+ to the first excited 2D state is the key factor for the transition-metal like chemistry of the alkaline earth atoms, perhaps even in the neutral molecule. This is, because the metals are usually bonded to more electronegative atoms, which make M to become like M^+ . This is a heuristic speculation, which shall be studied by us in forthcoming work.

2) Are the orbital interactions (and deformation densities) symmetry based (i.e. is ΔE_{orb1} strictly synonymous with eg, ΔE_{orb2} with t2g etc)?

Reply: Yes! The deformation densities stem from orbitals and densities, which have the respective symmetry.

REVIEWERS' COMMENTS:

Reviewer #2 (Remarks to the Author):

The authors have done all to convince me and I support publication of the revised manuscript.

I really do see the added value of the EDA-NOCV analysis with respect to the difference in binding between neutrals and cations. This was a very valuable addition to the manuscript.

I still have a hard time to use the 18 VE electron rule in this case, especially with the triplet ground state. But formally your arguments are right.

But in the end you do not need to overstate. Coordination Number 8 is very favorable for the heavier alkaline earth metals and also the orbital arrangements are favorable; both make the case for the stability of this structure in neutral and cation.

For my taste the reference to the 18 VE rule could be removed. But this is optional. Science is about content and not taste...

Response to referees:

I was very happy about the very positive remarks of the referee and the principle acceptance of our paper for publication in Nature Communications. We added a further paragraph on page 8 starting with "One referee noted.." where we address the remaining concern of the reviewer. We might not completely convince him on the value and validity of the 18 electron rule but I hope that our arguments are a step in this direction.